# Electrochemical Studies of the Corrosion Behavior of Al/SiC/PKSA Hybrid Composites in 3.5% NaCl Solution

Peter Ikubanni [1], Makanjuola Oki [1], Adekunle Adeleke [2], Olanrewaju Adesina [3], Peter Omoniyi [4,5,*] and Esther Akinlabi [6]

1 Department of Mechanical Engineering, College of Engineering, Landmark University, Omu-Aran 251101, Nigeria
2 Department of Mechanical Engineering, Faculty of Engineering, Nile University of Nigeria, Abuja 900001, Nigeria
3 Department of Mechanical Engineering, College of Engineering, Redeemers' University, Ede 232101, Nigeria
4 Department of Mechanical Engineering, University of Ilorin, Ilorin 240243, Nigeria
5 Department of Mechanical Engineering Science, University of Johannesburg, Johannesburg 2092, South Africa
6 Mechanical and Construction Engineering Department, Northumbria University, Newcastle NE1 8ST, UK
* Correspondence: omoniyi.po@unilorin.edu.ng or 219126794@student.uj.ac.za; Tel.: +27-622-635-779

**Abstract:** The corrosion behavior of metal matrix composites (MMCs) is accelerated by the inclusion of reinforcements. Hence, this study investigates the corrosion behavior of MMCs produced from Al 6063 matrix alloy with reinforcement particulates of silicon carbide (SiC) and palm kernel shell ash (PKSA) inclusion at different mix ratios. The MMCs were synthesized using the double stir casting technique. The corrosion behaviors of the composites in NaCl solutions were studied via gravimetric analysis and electrochemical measurements. The gravimetric analysis showed fluctuating dissolution rate of the samples in NaCl solution to indicate flawed film as well as corrosion product formation over the surface of the specimens. The observed corrosion mechanism of the samples was general and pitting corrosion. The presence of reinforcements within the Al6063 matrix acted as active sites for corrosion initiation. The range of values for $E_{corr}$ and $I_{corr}$ obtained in 3.5% NaCl at 24 h was between −220.62 and −899.46 mV and between 5.45 and 40.87 μA/cm$^2$, respectively, while at 72 h, the $E_{corr}$ values ranged from 255.88 to −887.28 mV, and the $I_{corr}$ ranged from 7.19 to 16.85 μA/cm$^2$. The Nyquist and Bode plots revealed the electrochemical corrosion behavior of the samples under investigation, with predominant reactions on the surface of the samples linked to charge transfer processes. The relative resistance to corrosion of the samples depends on the thin oxide film formed on the surface of the samples.

**Keywords:** palm kernel shell ash; aluminum matrix composite; silicon carbide; physicomechanical properties; microstructure

## 1. Introduction

The continuous progress witnessed in advanced engineering materials used in the automobile, marine, aviation, and building construction has been due to various scientific research breakthroughs [1–4]. Composite materials are a major component of advanced engineering materials and are categorized based on matrices [5–7]. Metal matrix composite (MMC) is one of the categories of composite materials in line with advanced engineering materials. Various metal matrices (Al, Mg, and Ti) have been used as base matrices. However, an aluminum matrix has been majorly researched [2,8,9]. Aluminum matrix composites (AMCs) are dexterous materials in mechanical engineering applications and sectors.

Recently, many efforts have been made to utilize AMCs because of their high specific strength and low thickness, high quality, good weight-to-strength ratio, and relatively high wear resistance [9–11]. The application of AMCs in automobiles and aviation has been attributed to their excellent physicomechanical characteristics [9,12]. Many monolithic

and hybrid reinforcements have been incorporated into aluminum matrices to produce AMCs [1,12–14]. The reinforcements used are either synthetic reinforcements (such as SiC, $Al_2O_3$, $B_4C$, and TiC) [15,16] or the combinations of two synthetic reinforcements or of synthetic reinforcement and reinforcements derived from the utilization of agro-residue such as rice husk ash (RHA), bean pod ash (BPA), corncob ash (CCA), and sugarcane bagasse ash (SBA) [11,12,17].

Many metallurgical routes are available in developing MMCs, including powder metallurgy, high-energy ball milling, stir casting, spray deposition, and squeeze casting [18,19]. However, due to the simplicity of operation, non-expensiveness, and relatively good uniform distribution of reinforcement particulates into the matrix, the stir casting method is mainly employed [2,9,18,20,21]. Although physicomechanical properties of MMCs can be enhanced through the introduction of reinforcement, corrosion characteristics can be significantly altered.

One of the germane parameters in assessing the application potential of AMCs is to consider their corrosion behaviors [1]. Extensive work has been carried out on physicomechanical and tribological characteristics of MMCs; however, further study is required for the corrosion behavior of hybrid reinforced MMCs, especially when agro-residue ashes are used as partial or total reinforcement. Reinforcements in MMCs initiate pits, especially at the matrix's secondary particles. This could be attributed to various chemical, electrochemical, or physical interactions between the matrix, reinforcements, and the simulated corrosion environment [1].

Nanjan and Murali [9] studied the effect of reinforcing Al6061 with graphite, $Al_2O_3$, SiC, and $B_4C$ at 4% and 5% using the stir casting route. The produced samples were immersed in 3.5% NaCl, and 1M HCl solutions and evaluations were carried out. The results revealed that there was good corrosion protection as well as good mechanical properties. The influence of hybrid RHA and $Al_2O_3$ particulates in equal proportions (1–5 wt.%) when incorporated in A356 alloy was studied under different ageing conditions. The samples produced via the stir casting route were immersed in 5% NaCl and 0.1 normal HCl solutions. The study revealed an enhancement in the composite's corrosion resistance [22].

The corrosion behavior of aluminum metal composites (AMCs) made with Al powder and SiC in an atmosphere with 3.5 wt.% NaCl was studied by Zakaria [1] at both ambient and high temperatures. Investigations were conducted into the effects of SiC reinforcement size and volume fraction on the microstructure and corrosion behavior of the manufactured MMCs. The corrosion resistance of Al/SiC MMCs was higher than that of the pure Al matrix. The Al/SiC corrosion rate decline was produced by shrinking SiC particles and increasing their volume percentage. Ononiwu et al. [23] conducted an electrochemical examination of composites made using Al-Si12 as a matrix, fly ash, and carbonized eggshell as reinforcements in 3.5 wt.% NaCl (2022). According to the investigation, as the weight fraction of the reinforcement rose, the corrosion rates of the cast AMCs increased. The wood particle was used as a reinforcement in the Al matrix in producing AMCs, and the corrosion properties were studied. The study revealed that the corrosion resistance of the unreinforced samples improved in 3.5 wt.% NaCl solution compared to the reinforced aluminum alloy [24]. The corrosion behavior of AMCs produced with hybrid reinforcement of SiC and rice husk ash (RHA) with Al6063 as the matrix was investigated by Haridass et al. [6]. A stir casting route was used to produce the AMCs, while the corrosion solution used was $AlCl_3$. The study revealed that the corrosion resistance of the hybrid reinforced AMCs increased with an increase in SiC and RHA addition. In addition, the hybrid aluminum composites exhibited higher corrosion resistance than the pure Al6063.

The corrosion behavior of various AMCs has led to different contradictory results due to the different matrices, reinforcements, and production techniques utilized. Hence, there is no consistent trend of scientific findings based on the stated conditions. Studies on the corrosion properties of AMCs produced using SiC and palm kernel shell ash (PKSA) are scarce in the literature. There is a lack of information on the influence of PKSA and SiC

particulates on the corrosion susceptibility of Al6063/SiC/PKSA composites. Therefore, the present study investigated the corrosion susceptibility of synthesized AMCs in 3.5 wt.% NaCl solution. The weight loss of the composites in both solutions was determined, and potentiodynamic polarization investigations were carried out.

## 2. Materials and Methods

### 2.1. Materials and Composite Production

Al 6063 served as the study's metal matrix, and silicon carbide (SiC) and palm kernel shell ash were used as reinforcements. (PKSA) Al6063 was acquired from an aluminum manufacturer in Lagos, Nigeria; SiC from a chemical supplier in Akure, Nigeria; and PKSA was acquired in accordance with the methods described by Ikubanni et al. [2]. The chemical composition of the Al6063 alloy was 0.48% magnesium, 0.43% silicon, 0.17% iron, 0.04% manganese with the rest being aluminum, while that of the PKSA was 66.9% $SiO_2$, 6.46% $Al_2O_3$, 5.72% $Fe_2O_3$, 5.52% CaO, 5.20% $K_2O$, 3.78% $P_2O_5$, 3.14% MgO, 0.53% $TiO_2$, and 2.75% loss on ignition. The SiC and PKSA reinforcements have average particle sizes of around 30 and 40 μm, respectively. The designation of the composites created is shown in Table 1. In accordance with numerous publications, the two-stir casting method was utilized as the production technique [5,13,25]. This technique is a liquid metallurgy route. The PKSA and SiC particulates required per composition in the matrix alloy were evaluated via the charge calculation method. Inherent moisture removal as well as improving wettability of the reinforcement in the matrix was performed by preheating at 250 °C. The matrix ingots were charged and melted in a gas-fired crucible at a temperature above the liquidus temperature of the matrix alloy (750 °C). This was to ensure complete melting of the matrix alloy. The preheated reinforcement particulates were later charged into molten matrix alloy that was in semi-solid state, and the slurry was manually stirred for between 5 and 10 min. The slurry was later superheated to 800 °C and mechanically stirred the second time at 400 rpm for 10 min. At the final stage, the slurry was discharged into a prepared sand mold and allowed to solidify. The composite product was then obtained after solidification.

**Table 1.** Designation of composite samples produced.

| Sample Designation | Aluminum (6063) (wt.%) | PKSA (wt.%) | SiC (wt.%) |
|:---:|:---:|:---:|:---:|
| A0 | 100 | 0 | 0 |
| A1 | 98 | 0 | 2 |
| A5 | 90 | 8 | 2 |
| A6 | 98 | 2 | 0 |
| A9 | 90 | 2 | 8 |

### 2.2. Preparation of Specimen

The sample from each composition in Table 2 was cut to $\varnothing 30 \times 3$ mm dimensions. Each sample was ground using different grit sizes between 240 and 1000 SiC paper before being cleaned with acetone, then washed with water and dried. The weight of the sample was measured with a digital electronic balance, and the thickness of the sample was obtained using a vernier gauge. The microstructures of the samples were examined using a Vega 3 TESCAN model scanning electron microscope.

### 2.3. Physical and Mechanical Properties

The density and porosity, which are the physical properties of the composites, were determined as described by Prasad et al. [26]. However, Brinell's hardness value of the composites was determined using ASTM E10-18 standard [27]. The average hardness value of the four indents made on each sample for data reliability purposes was used. The tensile test was performed using a 30 mm gauge length and 5 mm diameter sample based on the ASTME8/E8M-16 standard [28]. A universal testing machine (UTM) (Model No: Instron 3369) at $10^{-3}$/s strain rate was used. The data curation was carried out in triplicate for data reliability.

**Table 2.** Physical and mechanical properties of the fabricated samples.

| Sample ID | Theoretical Density (g/cm$^3$) | Porosity (%) | Hardness (BHN) | YS (MPa) | UTS (MPa) | % Elong. |
|---|---|---|---|---|---|---|
| A0 | 2.70 | 2.063 | 73.019 | 78.837 | 116.087 | 7.6 |
| A1 | 2.71 | 2.177 | 75.792 | 84.469 | 124.470 | 7.2 |
| A5 | 2.56 | 2.289 | 80.550 | 93.419 | 128.169 | 5.8 |
| A6 | 2.66 | 2.128 | 74.093 | 81.402 | 123.402 | 7.4 |
| A9 | 2.70 | 2.074 | 85.488 | 101.962 | 133.462 | 5.0 |

YS—Yield Strength; UTS—Ultimate Tensile Strength; %Elong.—Percentage Elongation.

### 2.4. Corrosion Test

2.4.1. Immersion Test/Gravimetric Measurement

Using a digital scale with 0.1 mg accuracy, weight loss was recorded to assess the corrosion rate of the hybrid Al composites. The corrosive environment was created as a 3.5 wt.% NaCl solution. Prior to being submerged into the two solutions individually, each specimen was weighed. Before reweighing, the samples were taken out of the solutions, cleaned, and dried ($W_A$). This procedure was carried out for 15 days at room temperature.

The corrosion rates given in millimeter penetration per year (mm/year) and the weight loss rates were calculated. Photographs were used to inspect the deteriorated surfaces. A glass cover was placed on top of the jar for the samples in the elevated temperature corrosion test to stop evaporation. Using Equation (1) based on ASTM G31-12a [29], the weight loss measured was calculated and, using Equation (2), the corrosion rate was assessed.

$$\text{Weight loss, } W = W_B - W_A \tag{1}$$

$$\text{Corrosion rate, } CR = \frac{K \cdot W}{A \cdot D \cdot T} \tag{2}$$

where CR is the corrosion rate (mm/year), $K$ is a constant ($8.766 \times 10^4$), $T$ is the time of exposure (h), $A$ is the area (cm$^2$), $W$ is the weight loss (mg), and $D$ is the density of the material (g/cm$^3$).

2.4.2. Electrochemical Corrosion Test

The manufactured composite was subjected to accelerated electrochemical investigations utilizing the potentiodynamic technique. The 3.5 wt.% NaCl solution at 25 °C was used for the potentiodynamic measurements. Prior to putting the samples in the measuring vessel, they were polished and cleaned to a surface area of 1 cm$^2$. The composite sample served as the working electrode, while platinum served as the counter electrode and silver/silver chloride (Ag/AgCl) serves as the reference electrode. To reach the requisite corrosion potential ($E_{corr}$) for the experiment, the apparatus was set for around 30 min. Then, an automated potential shift was applied at a rate of $10^{-3}$ V to record the anodic polarization curves. In accordance with ASTM G102-89 [30], the polarization measurements were performed from −1.5 to +1.5 V at a scan rate of 0.0016 V/s. The measurements were performed at 24 and 72 h of immersion of the various specimens in 3.5 wt.% NaCl solution. The corrosion current densities ($I_{corr}$) and the corrosion potential ($E_{corr}$) for the various samples were calculated from the Tafel plots of log current versus potential. The electrochemical impedance spectrometry (EIS) measurement was obtained using the potentiodynamic apparatus, which was later employed to obtain the Nyquist and Bode plots.

### 3. Results and Discussion

### 3.1. Microstructure of Produced Composite

Figure 1a reveals the microstructure of the unreinforced matrix metal. There is no intermetallic grain observed in the microstructure. The microstructures of some other

representative samples are displayed in Figure 1b–e. The microstructures of the monolithic reinforced sample of SiC and PKSA are represented as Figure 1b,d, respectively. However, the microstructures of the representative samples for the hybrid reinforced composites are shown in Figure 1c,e. It can be observed that the reinforcement particulates were uniformly distributed in the matrix. The indication of crack formation as well as enlarged pores was not observed in the composites produced. Hence, there is an indication of good wetting condition between the reinforcement particulates and the matrix as a result of the selected processing techniques and parameters used in the composite synthesis [31,32]. The uniformly distributed reinforcement particulates revealed by the micrographs confirm the reliability of the double stir casting technique in breaking the surface tension between the composite components (the reinforcement particulates and the matrix alloy) [5]. Hence, entrapped air bubbles in the slurry are allowed to escape during the double stir casting processes.

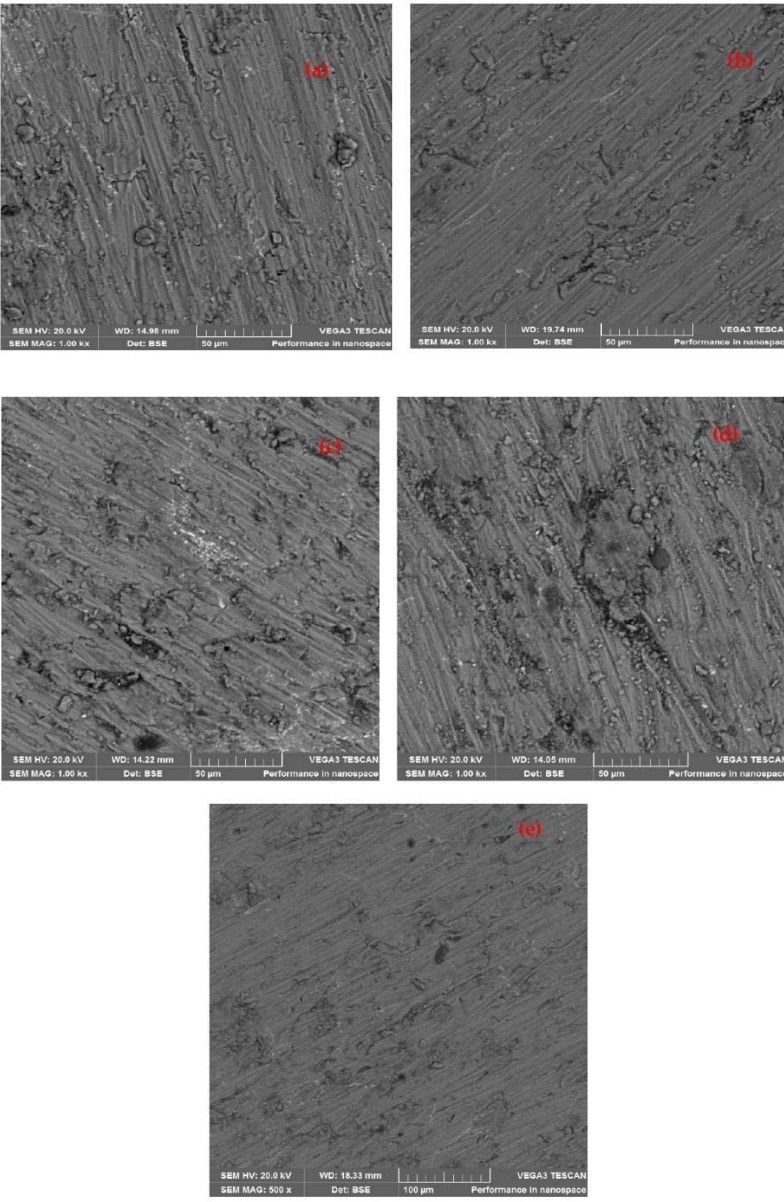

**Figure 1.** Scanning electron micrographs of (**a**) sample A0, (**b**) sample A1, (**c**) sample A5, (**d**) sample A6, (**e**) sample A9.

### 3.2. Physical and Mechanical Properties

The physical properties (theoretical density and percentage porosity) and mechanical properties (hardness, yield strength (YS), ultimate tensile strength (UTS), and percentage elongation (% El.)) are presented in Table 2. The theoretical density and percentage porosity of the unreinforced alloy were 2.7 g/cm$^3$ and 2.063%, respectively. Sample A5 gave the lowest density (2.56 g/cm$^3$), followed by sample A6 with 2.66 g/cm$^3$, while sample A1 gave the highest density (2.71 g/cm$^3$). The reason for the low density observed in samples A5 and A6 was due to the quantity of PKSA particulates in the matrix. The density of PKSA is lower compared to SiC and matrix alloy. Hence, its presence reduces the overall density of the produced composite. However, the density of sample A1 was higher than that of sample A0 because the density of SiC was higher compared to the matrix. The presence of more SiC particulates (8%) than PKSA particulates (2%) in sample A9 increased the density of the composite. Most biomass reinforcement particulates are known to have lower densities, which invariably affect the overall density of the composite [5].

All the samples show percentage porosities that are lower than 2.4%, a value showing that the produced composites are within the permissible limits for cast MMCs [32]. The obtained porosity percentage values re-affirmed the reliability of the processing route used in ensuring efficient surface tension breakage between the matrix and the reinforcement particulates.

The hardness value of any material has great influence on the material strength, toughness, and wear resistance. It can be observed from Table 3 that Brinell's hardness value, the YS and the UTS followed a similar pattern. The unreinforced sample (sample A0) has a hardness value of 73.02 BHN, a YS of 78.84 MPa, and a UTS of 116.09 MPa. Sample A1 reinforced with 2% SiC has values of 75.79 BHN, 84.47 MPa, and 124.47 MPa for hardness value, YS, and UTS, respectively. This was possible because of the hardness of SiC, which is greater than that of the matrix alloy. PKSA particulates contain hard phases of silica and other strengtheners; however, the small amount of PKSA (2%) in monolithic reinforced sample A6 resulted in a lower hardness value obtained compared to sample A1 but higher than sample A0. The combine reinforcement particulates of SiC and PKSA gave much better hardness values. However, sample A9 (2% PKSA and 8% SiC) gave the highest hardness value. The presence of some intermetallics has the ability to improve the hardness of the composites. The rise in the volume of precipitated phases or high dislocation density suggests an increase in hardness. Hence, the matrix grain size increases along with the surface area of the added reinforcement particulates [33,34].

**Table 3.** Corrosion parameters from PDP curves for samples in NaCl solution.

| Sample Immersed in 3.5% wt. NaCl Solution for 24 h | | |
| --- | --- | --- |
| **Sample** | **$E_{corr}$ (mV)** | **$I_{corr}$ ($\mu$A/cm$^2$)** |
| A0 | −220.619 | 5.450 |
| A1 | −760.907 | 40.868 |
| A5 | −884.590 | 11.798 |
| A6 | −862.798 | 9.971 |
| A9 | −899.467 | 11.524 |
| **Sample Immersed in 3.5% wt. NaCl Solution for 72 h** | | |
| **Sample** | **$E_{corr}$ (mV)** | **$I_{corr}$ ($\mu$A/cm$^2$)** |
| A0 | −255.875 | 7.370 |
| A1 | −543.928 | 16.850 |
| A5 | −856.120 | 6.810 |
| A6 | −846.725 | 7.190 |
| A9 | −887.281 | 7.200 |

Since the YS and UTS results followed the sample pattern with the hardness value, the presence of the hard nature of silica and other strengthening oxides in the PKSA particulates in the samples (A5, A6, and A9) led to improved YS and UTS compared to the

unreinforced matrix. However, comparing the monolithic reinforced samples A1 and A6 with 2% SiC and 2% PKSA, respectively, sample A1 revealed higher YS and UTS because SiC has better strength than PKSA. YS and UTS improvement in MMCS can be linked to three mechanisms: the grain boundary strengthening known as the Hall–Petch effect, the mismatch of the coefficient of thermal expansion (CTE) of the matrix and reinforcement, as well as the Orowan strengthening mechanisms [35].

The % elongation of the samples produced was observed to be between 5.0 and 7.6%. The unreinforced alloy (sample A0) showed the best % elongation. Sample A9 has the lowest % elongation of 5.0%, which is attributed to the plastic strain sustenance ability as a result of the hard constituents present in the PKSA and the SiC particulates. Hence, a reduction in the ductile phase of the matrix as well as decrease in percentage elongation of the composite were observed [10,21].

### 3.3. Gravimetric Analysis

Corrosion Behavior in 3.5 wt.% NaCl Environment

Figures 2 and 3 show the weight changes and corrosion rates against the exposure time of the samples in a 3.5% salt solution. The result fluctuations show weight losses and weight gains. The weight gain could be attributed to the formation of adherent corrosion products on the surfaces of the samples, which may lead to the temporary reduction in an area of the substrate exposed to the corrosives. However, the weight loss could be due to the breakdown of the flawed films cumulating in the gradual removal of reinforcements within several days.

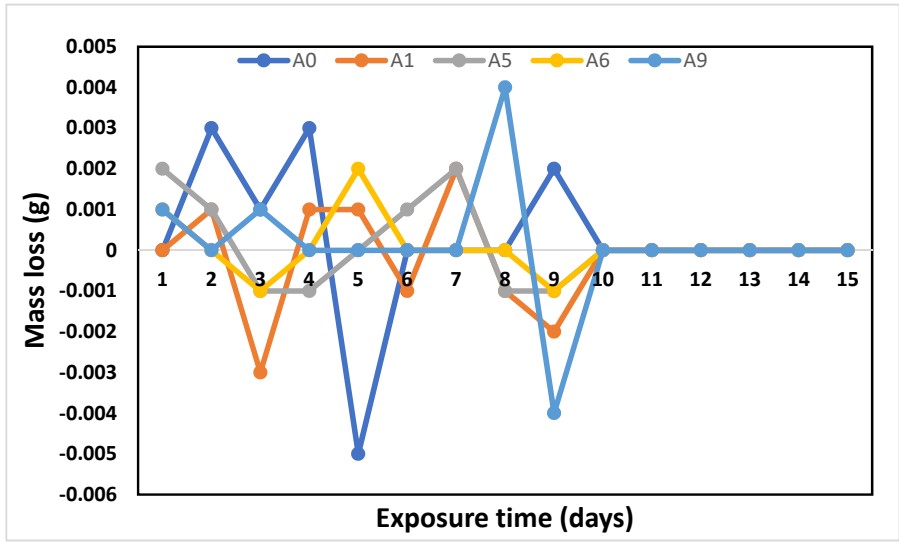

**Figure 2.** Mass changes of specimens against the time of exposure in 3.5 wt.% NaCl solution.

The mass loss in the 3.5 wt.% NaCl solution is 0.04 g. Since the composites are almost stable with some resistance to general corrosion in 3.5 wt.% NaCl, it may be suitable for usage in saline or marine environments with further protection. This is in tandem with the study of Alaneme and Bodunrin [36] where alumina reinforced Al6063 matrix was immersed in 3.5 wt.% NaCl, and similar results with the present study were obtained. The influence of the reinforcement addition on the corrosion behavior was slightly different from the unreinforced matrix, as the level of corrosion rate was less than 0.1 mmpy. There is an invariant corrosion behavior irrespective of the percentage weight of the SiC and PKSA in the composites. However, in the study of Alaneme et al. [37], it was reported that monolithic (SiC or GSA) reinforced samples showed better corrosion resistance than the hybrid (SiC and GSA) reinforced samples and unreinforced samples. The mass gain during gravimetric measurements of the samples may make the materials suitable for use in marine environments. PKSA contains silica, which hampers $Al_4C_3$ phase formation due

to SiC and Al interfacial reaction, extending the matrix's susceptibility to corrosion [37]. This is because silica crucibles are not used for the melting of the aluminum. SiC does not react with Al; however, the formation of intermetallic phases which serve as initiation points for corrosion is possible. In addition, Al-SiC composite is based on the absence of the reaction.

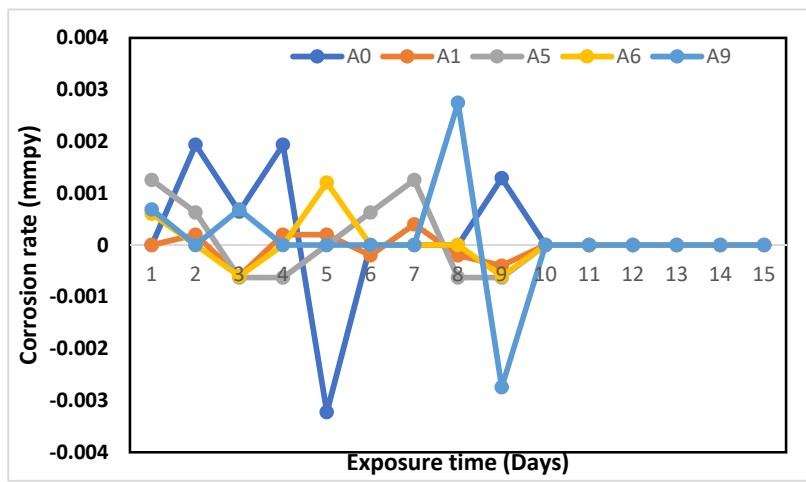

**Figure 3.** Corrosion rate of specimens versus the time of exposure in 3.5 wt.% NaCl solution.

Figure 4a,b show the obtained surface of the samples after exposure to NaCl solution for 15 days compared with the sample not immersed in the solution (Figure 4c). The corrosion behaviors of all the specimens are similar, with surges in the positive and negative directions reminiscent of observations made during potential measurements. Such surges infer some crack and heal events occurring at the bases of all the pits. The corrosion behaviors of all the specimens are similar, with surges in the positive and negative directions reminiscent of observations made during potential measurements. Such surges infer some crack and heal events occurring at the bases of all the pits.

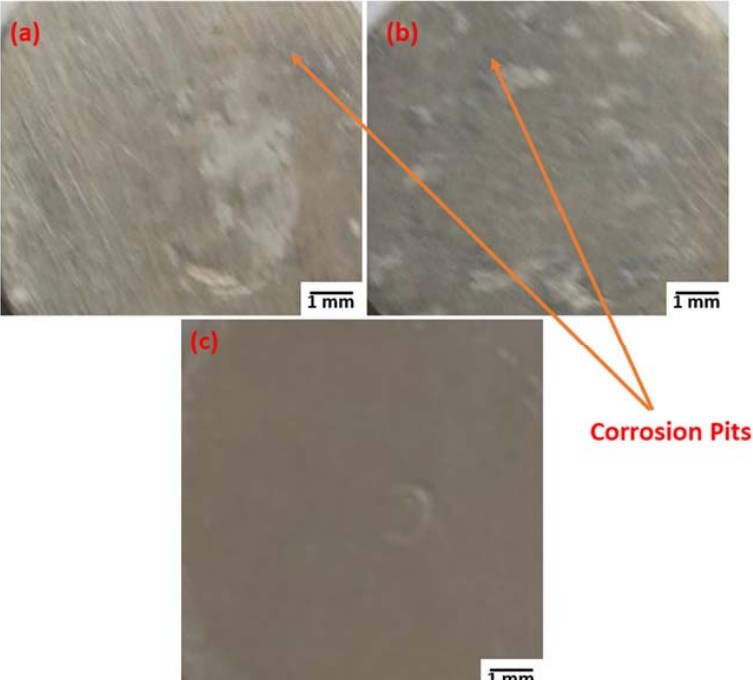

**Figure 4.** Corroded samples of (**a**) A9 and (**b**) A1 after immersion in NaCl solution for 15 days (**c**) uncorroded sample.

The 3.5 wt.% NaCl solution (pH = 7.4) had very little pitting corrosion on the composites, though the pH level could be to blame for these findings. Four (4) steps of pitting corrosion are initiated by Cl ions. The hydrolysis of corrosion products is invariably combined with metal oxidation in the pitting corrosion mechanism, producing localized acidity. The substantial separation of the cathodic half-reaction and the anodic half-reaction helps to maintain this localized acidity. The electrolyte gradually becomes more acidic as a result of insufficient oxygen penetration. Anion electrons migrate toward the created pit due to the acidity. The side products created during the corrosion process frequently fill the pits on the metal surface.

The cathode is the surrounding surface that has been passivated, whereas anodic processes start on the metal surface that is exposed to the electrolyte. As a result, the particles from the second phase, the reinforcing inclusions, show up on the metal surface. As local anodes, the particles that precipitate along grain boundaries may cause localized galvanic corrosion and the emergence of early pits. Localized strains that manifest as dislocations on the surface have the potential to become anodes and start pits. For example, when exposed to a sodium chloride solution high in oxygen, the metal surface functions as a cathode and the pits that form as anodes. The electrolyte's chloride anions are eventually drawn to the excess positive charge created by the metal cations being produced in the pit. The ensuing molecules of the metal chloride now react with the water in the environment producing metal hydroxide and hydrochloric acid, further speeding up the corrosion rate.

### 3.4. Electrochemical Measurement of the Samples

Corrosion Behavior in 3.5 wt.% NaCl Solution Using PDP

The potentiodynamic polarization curves of the composites exposed to 3.5% wt. NaCl solution for 24 and 72 h are shown in Figure 5a,b, respectively. The $E_{corr}$ obtained is −0.22 V, which is nobler than those of reinforced specimens which are −0.761 V for A1, −0.885 V for A5, −0.863 V for A6, and −0.899 V for A9. The behaviors of the samples in NaCl indicated their susceptibility to corrosion attacks. This is an affirmation of what was observed during the gravimetric experiment. In 3.5% NaCl, the $E_{corr}$ obtained was −220.62 mV. Furthermore, the corrosion current and potential shown in Figure 5a for the reinforced specimens were relatively low, with $I_{corr}$ between 5.45 and 40.87 μA/cm$^2$ and $E_{corr}$ between −220.62 and −899.48 mV. This phenomenon may be due to chloride attack at the aluminum–reinforcement interface.

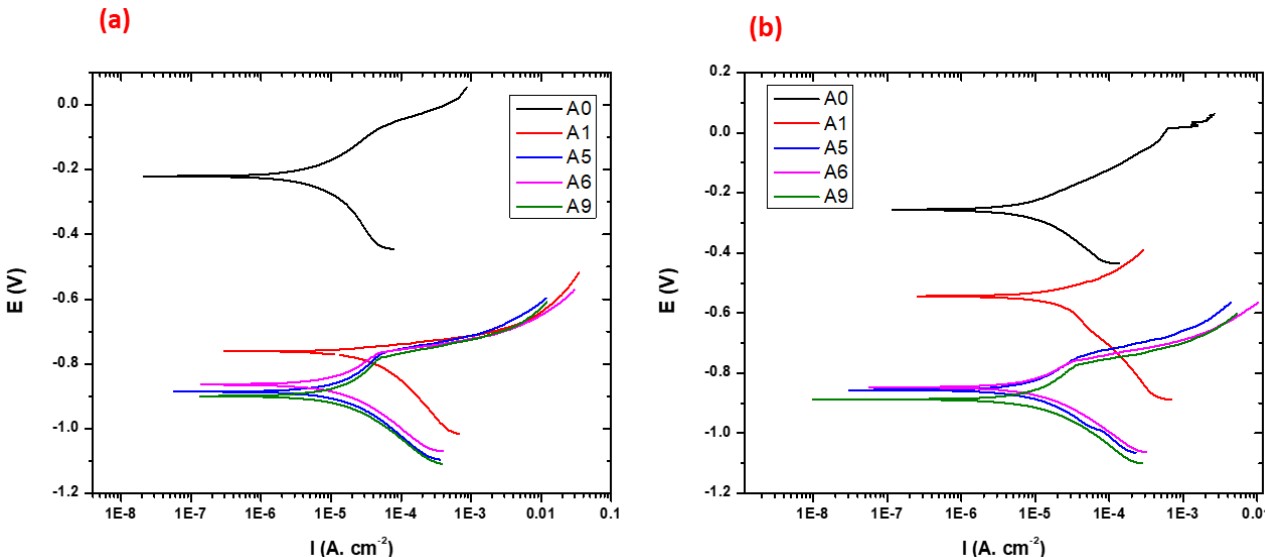

**Figure 5.** Polarization curves for the composites in 3.5% wt. NaCl solution at (**a**) 24 and (**b**) 72 h.

Table 3 shows $I_{corr}$ and $E_{corr}$ values for Samples A0, A1, A5, A6, and A9. At 24 h immersion in NaCl solution, it can be observed that sample A0 had the lowest $I_{corr}$ value

(5.45 µA/cm$^2$), while at 72 h immersion time in NaCl solution, sample A5 gave the lowest I$_{corr}$ value (6.809 µA/cm$^2$). A low I$_{corr}$ value indicates a low corrosion rate in NaCl solution. The unreinforced alloy showed an increase in I$_{corr}$ values from when it was immersed in NaCl solution at 24 (5.45 µA/cm$^2$) to 72 h (7.373 µA/cm$^2$). This implies expectedly that the corrosion rate of the unreinforced alloy increased with exposure time in NaCl. The I$_{corr}$ values of other samples immersed in NaCl solution increased at 72 h of exposure. However, the rate of increase in I$_{corr}$ for the composites was lower than for the unreinforced alloy, and as discussed earlier, this is related to the potential pitting passivation window, which varies with exposure regimes, $\Delta E = (E_{pp} - E_{corr}) \geq 100$ mV. The summary of the E$_{corr}$ and I$_{corr}$ values of the samples is shown in Table 3.

Figure 5a,b reveal that irrespective of the exposure periods of the sample in the NaCl environment, the unreinforced sample A0 showed higher corrosion resistance than other samples. In addition, the sample reinforced with 2% SiC (sample A1) showed better corrosion resistance than samples A5, A6, and A9. The high corrosion resistance of sample A1 to other reinforced samples is due to the small amount of SiC reinforcement introduced into the matrix. Sample A6 with 2% PKSA inclusion in its matrix showed better corrosion resistance compared to samples A5 and A9. However, the obtained value is far lower than the value achieved in sample A1. The presence of some metallic oxides in the PKSA included in the matrix is a point of initiation of corrosion attacks in the NaCl environment. However, the hybrid reinforced composites of sample A5 (2% SiC and 8% PKSA) and sample A9 (2% PKSA and 8% SiC) showed the least corrosion resistance in the subjected environment. This implied that 10% of the hybrid reinforced replacement in the Al matrix served as very strong active sites for pitting initiation.

### 3.5. EIS Spectra of the Samples

The Bode plots of the samples immersed in 3.5% NaCl solution for 24 and 72 h are shown in Figures 6 and 7, respectively. The Bode profile for the impedance modulus against frequency showed the samples' similar patterns. The unreinforced sample A0 has the highest impedance (Figure 6a). The reinforced composites could not quickly provide a protective film layer in the corrosive media like the unreinforced sample at 24 h immersion time. However, other samples showed better impedance modulus at 72 h immersion time in NaCl solution. It can be observed from Figure 7a that there was a steady rate of impedance modulus of the reinforced alloy in the NaCl solution with respect to the frequency values. This implied that the unreinforced sample may have shown better corrosion resistance, unlike the samples with reinforcement inclusions.

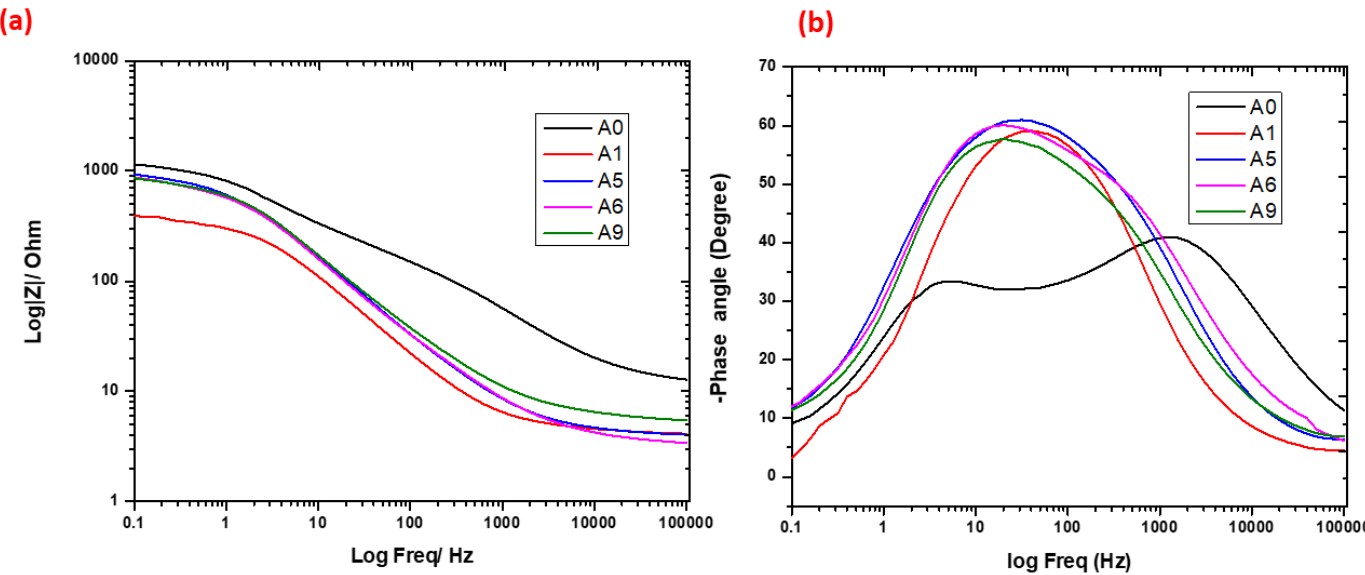

**Figure 6.** (**a**) Bode profiles, (**b**) Bode phase profiles of samples in 3.5% NaCl for 24 h.

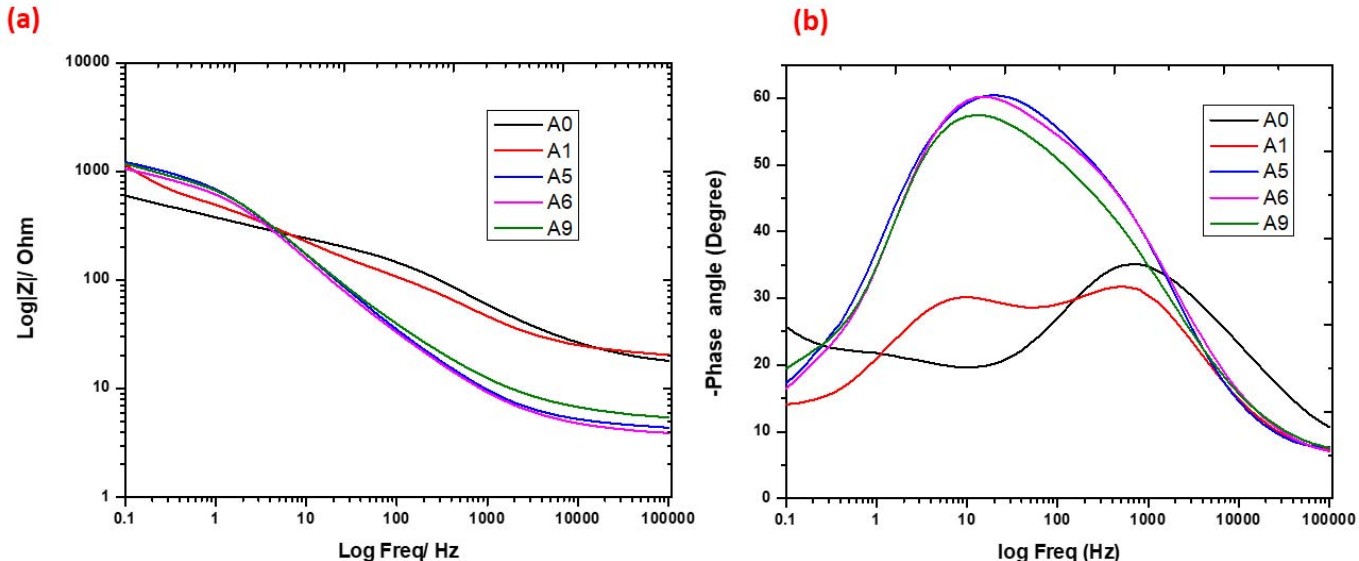

**Figure 7.** (**a**) Bode profiles, (**b**) Bode phase profiles of samples in 3.5% NaCl for 72 h.

Table 4 shows the samples' EIS fitting outcomes for solutions containing 3.5 wt.% NaCl. $R_{ct}$ stands for the samples' corrosion resistance in the solution. The thin oxide film that forms on the sample surfaces determines how resistant the samples are to corrosion. This could have restricted ionic and electronic conductivity and slowed down electrochemical reactions on the samples' surfaces [38].

**Table 4.** Fitting parameters for EIS spectra of samples in 3.5% NaCl solution at 24 and 72 h.

| Sample | Time (h) | Rs ($\Omega\cdot cm^2$) | Rf ($\Omega\cdot cm^2$) | Y01 ($\mu F\cdot cm^2$) | n1 | $R_{ct}$ ($\Omega\cdot cm^2$) | Y02 ($\mu F\cdot cm^2$) | n2 |
|---|---|---|---|---|---|---|---|---|
| A0 | 24 | 11.06 | 236.00 | 59.3 | 0.653 | 1010 | 199 | 0.69 |
|    | 72 | 15.72 | 66.37 | 101 | 0.594 | 247.5 | 1360 | 0.948 |
| A1 | 24 | 4.046 | 3.012 | 288 | 0.732 | 382.6 | 53.2 | 0.886 |
|    | 72 | 19.55 | 125.9 | 93.0 | 0.531 | 979.7 | 545 | 0.647 |
| A5 | 24 | 3.729 | 1.537 | 77.0 | 0.768 | 995.3 | 180 | 0.736 |
|    | 72 | 4.387 | 102.9 | 236 | 0.727 | 1080 | 15.3 | 0.979 |
| A6 | 24 | 3.358 | 33.37 | 209 | 0.736 | 880.4 | 54.5 | 0.802 |
|    | 72 | 3.914 | 38.93 | 198 | 0.741 | 952.5 | 57.5 | 0.832 |
| A9 | 24 | 5.299 | 7.481 | 115 | 0.751 | 903.1 | 136 | 0.725 |
|    | 72 | 5.43 | 4.51 | 189 | 0.714 | 1040 | 600 | 0.829 |

For the samples immersed in the NaCl solution (Table 4), it can be observed that samples A1, A5, A6, and A9 showed increased $R_{ct}$ values in the 3.5% NaCl solution from 24 to 72 h immersion time. The $R_{ct}$ values at 24 h immersion time for A1, A5, A6, and A9 were 382.6, 995.3, 880.4, and 903.1 $\Omega\cdot cm^2$ and increased to 979.7, 1080, 952.5, and 1040 $\Omega\cdot cm^2$ at 72 h immersion time, respectively. The increase in the $R_{ct}$ values can be attributed to the decrease in the oxygen content present in the experimental setup as well as the quick formation of flawed oxide layers of corrosion products on the surface of the samples, which could impede further action of the solutions on the samples. The passive film resistance $R_f$ values showed an initial decrease for sample A0 from 236 to 66.37 $\Omega\cdot cm^2$, while samples A1, A5, and A6 revealed increased $R_f$ values at 24 to 72 h immersion time in 3.5% NaCl solution (Table 4).

## 4. Conclusions

The influence of SiC and PKSA reinforcements in Al6063 was preliminarily studied. The gravimetric analysis showed the presence of corrosion product formation with minimal pits. The corrosion mechanism of the samples immersed in NaCl solution was general and pitting corrosion. The composites displayed similar electrochemical behaviors in the 3.5% NaCl solution. Sample A0 (unreinforced alloy) initially had better corrosion resistance when compared with the reinforced samples, but its corrosion resistance decreased with time. The presence of additives or reinforcements within the matrix of Al6063 acted as active sites for corrosion initiation. The different chemical properties of the reinforcement inclusions in Al6063 are likely to form discontinuous/flawed oxide layers wherever they intersect surfaces of the composites, resulting in stress raisers prone to pitting corrosion initiation sites. In conclusion, the composites will be utilized in areas less prone to corrosion attack with advanced protection in application areas.

**Author Contributions:** Conceptualization, P.I., M.O. and A.A.; methodology, P.I., M.O., A.A. and O.A.; software, P.I., M.O., A.A. and O.A.; validation, P.O., E.A. and M.O.; formal analysis, P.O. and E.A.; investigation, P.I. and A.A.; resources, P.I., M.O., A.A., O.A., P.O. and E.A.; data curation, P.I., M.O., O.A. and A.A.; writing—original draft preparation, P.I., M.O., A.A. and O.A.; writing—review and editing, M.O., A.A., P.O. and E.A..; visualization, P.O., E.A. and A.A.; supervision, M.O., E.A., A.A. and O.A.; project administration, M.O. and E.A.; funding acquisition, E.A.. All authors have read and agreed to the published version of the manuscript.

**Funding:** This research received no external funding.

**Data Availability Statement:** The data presented in this study are available on request from the corresponding author.

**Acknowledgments:** We thank Landmark University's administration for providing the ideal setting for conducting this study.

**Conflicts of Interest:** The authors declare no conflict of interest.

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
