# Peer review of "Electrochemical Studies of the Corrosion Behavior of Al/SiC/PKSA Hybrid Composites in 3.5% NaCl Solution"

_jcs, doi:10.3390/jcs6100286_

Round 1

Reviewer 1 Report (Previous Reviewer 1)

The manuscript is improved, considering most of the reviewers' comments.

Author Response

Thank you for your response.

Reviewer 2 Report (Previous Reviewer 2)

in attached file

Author Response

This manuscript is a resubmission of an earlier submission. The following is a list of the peer review reports and author responses from that submission.

Round 1

Reviewer 1 Report

The manuscript reports a piece of corrosion work of several particle reinforced aluminium samples A0 to A9. The general conclusion that the particles activate corrosion is not new. It is more interesting to find out how different reinforcement, PKSR and SiC, and their concentrations, affect the corrosion behaviour, but authors did not perform detailed investigation and analysis in this line. Other comments are as follows:

(1) The microstructural features of the test samples should be given, such as grain size, particle distribution/agglomeration etc.  Corrosion behaviour is related to microstructures.

(2) Mechanical properties of the samples should also be given, like hardness and tensile strength.

(3) Equation 2 seems incorrect: Why K has that value? What is T? IE does not appear in the equation.

(4) Line 157: 10-3 V? What is this? It does not match the scan rate of 0.0016V/s.

(5) No EIS measurement details are given.

(6) Figure 1 and 2 do not show difference between the samples. This normally means the test condition could not distinguish the different behaviour of the samples. 

(7) Figure 3: the images are of poor resolution. reinforcement particles and pits cannot be seen clearly. They do not serve any purpose with the current quality.

(8) Line 213-223: The discussion here is simply speculation. You need to provide evidences for your samples by detailed examination of the corroded surfaces with high resolutions.

(9) Section 3.2.1: testing the samples after immersion for 24 h and 72 h, you were testing the corroded surface of the samples and the stability of the corrosion film. What happen to the bare surface? additional tests were required.

(10) Line 230: Did you do tests in H2SO4?

(11) Figure 11 and Table 4: There are differences between different samples. These need to be discussed. Should include high resolution images to show corrosion pits and morphology of pits of different samples. To advance knowledge, it is important to find the effect of different reinforcements and their concentrations.

(12) Line 295: It is necessary to show evidence of "discontinuous/flawed oxide layers ...." in your samples, not just guess. 

Reviewer 2 Report

in attached file

Round 2

Reviewer 1 Report

Authors have not modified the manuscript according to reviewers' comments. The revised version is almost the same as the original version. Very few modifications have been done in the revised version. 

Reviewer 2 Report

in an attached file
